# Closing the Gap in Surveillance and Audit of Invasive Mold Diseases for Antifungal Stewardship Using Machine Learning

**DOI:** 10.3390/jcm8091390

**Published:** 2019-09-05

**Authors:** Diva Baggio, Trisha Peel, Anton Y. Peleg, Sharon Avery, Madhurima Prayaga, Michelle Foo, Gholamreza Haffari, Ming Liu, Christoph Bergmeir, Michelle Ananda-Rajah

**Affiliations:** 1Department of Infectious Diseases, The Alfred Hospital and Central Clinical School, Monash University, Melbourne 3004, VIC, Australia; 2General Medical Unit, Alfred Health, Melbourne 3004, VIC, Australia; 3Infection and Immunity Program, Monash Biomedicine Discovery Institute, Department of Microbiology, Monash University, Clayton 3800, VIC, Australia; 4Haematology and Bone Marrow Transplant Service, Alfred Health, Melbourne 3004, VIC, Australia; 5Faculty of Information Technology, Monash University, Clayton 3800, VIC, Australia

**Keywords:** invasive fungal diseases, antifungal stewardship, machine learning, natural language processing, aspergillosis, mold infections

## Abstract

Clinical audit of invasive mold disease (IMD) in hematology patients is inefficient due to the difficulties of case finding. This results in antifungal stewardship (AFS) programs preferentially reporting drug cost and consumption rather than measures that actually reflect quality of care. We used machine learning-based natural language processing (NLP) to non-selectively screen chest tomography (CT) reports for pulmonary IMD, verified by clinical review against international definitions and benchmarked against key AFS measures. NLP screened 3014 reports from 1 September 2008 to 31 December 2017, generating 784 positives that after review, identified 205 IMD episodes (44% probable-proven) in 185 patients from 50,303 admissions. Breakthrough-probable/proven-IMD on antifungal prophylaxis accounted for 60% of episodes with serum monitoring of voriconazole or posaconazole in the 2 weeks prior performed in only 53% and 69% of episodes, respectively. Fiberoptic bronchoscopy within 2 days of CT scan occurred in only 54% of episodes. The average turnaround of send-away bronchoalveolar galactomannan of 12 days (range 7–22) was associated with high empiric liposomal amphotericin consumption. A random audit of 10% negative reports revealed two clinically significant misses (0.9%, 2/223). This is the first successful use of applied machine learning for institutional IMD surveillance across an entire hematology population describing process and outcome measures relevant to AFS. Compared to current methods of clinical audit, semi-automated surveillance using NLP is more efficient and inclusive by avoiding restrictions based on any underlying hematologic condition, and has the added advantage of being potentially scalable.

## 1. Introduction

Antifungal stewardship (AFS) is a growing area of clinical importance in hospitals worldwide that manage patients vulnerable to invasive fungal diseases (IFD) [1,2]. Like antibiotics, overuse of antifungal drugs is common but largely driven by the high morbidity and mortality of IFD [3] combined with insensitive diagnostic tools [4,5]. A recent systematic review showed that the effectiveness of most AFS programs is measured by savings in antifungal drug cost and consumption [6], a secondary consideration of AFS [7], rather than by indicators that actually reflect good quality of care such clinical outcomes, effectiveness of antifungal prophylaxis and processes including therapeutic drug monitoring, and speed of the diagnostic work-up [1,6,7]. Most AFS interventions preferentially target high-risk patients such as hemopoietic stem cell transplant (HSCT) recipients and acute leukemia patients as well as *Candida* infections which are easier to monitor than the more common [3,8,9] but diagnostically challenging invasive mold diseases (IMD) like Aspergillosis [6].

Surveillance, audit and feedback are fundamental to antimicrobial stewardship and facilitate the implementation, assessment and benchmarking of these programs. Whilst such mechanisms are well established for antibiotics [10], they do not exist for AFS. Recent British guidelines cite surveillance and audit as the first step in AFS in order to identify gaps in practice [1]. Similar views from low income settings like India recognize that regular audit and feedback of local (i.e., institutional) epidemiology acquired through continuous surveillance best drive “corrective and preventive actions” in AFS practice [2].

Surveillance and audit of IMD, is inefficient and laborious whether undertaken retrospectively [8,9,11,12,13,14,15] or less often, prospectively [3,16,17]. The largest prospective fungal surveillance study by the TRANSNET consortium (2001–2006) was characterized by high operational costs, targeted only HSCT recipients rather than all hematology patients, and had a finite timeline with the high cost of case finding and data collection posing a major barrier [3]. Retrospective studies have usually been restricted to specific risk groups [8,9,11,12] or individual antifungal agents [9,13,14] due to an inability to perform population scale surveillance efficiently.

In response to the difficulties of IMD surveillance we developed natural language processing (NLP) of chest computed tomography (CT) reports [18,19,20] to facilitate surveillance of IMD. Chest CT was targeted for screening because it is the cornerstone of IMD diagnosis: IMD usually presents as culture-negative pneumonia in 90–100% of cases [3,16,17]; CT is the first investigation performed when IMD is suspected; it is widely available and performed with a higher frequency than invasive tests like fiberoptic bronchoscopy. Although the radiographic features of IMD are non-specific, they are highly persuasive in the appropriate clinical context [21] even when results of microbiological investigations are negative [22]. It should be emphasized that NLP is not a diagnostic tool, but rather a computational method of analyzing radiologist language for features of fungal pneumonia that is useful for disease classification [18,19,20].

The NLP model used here [18,19] is part of a platform technology called fungalAi^TM^ (www.fungalAi.com) that also includes deep learning-based image recognition for fungal pneumonia on chest CT in development [23] and an expert system that integrates antifungal drug and microbiology data [24] to improve fungal prediction. Technical descriptions of the development and internal validation of the statistical NLP model used here [18,19] have been published. The aim of this study was to characterize institutional IMD burden along with process and outcome measures relevant to AFS in order to identify areas for improvement. What follows is a clinical (rather than technical) study of applied machine learning that represents an *in silico* analysis of NLP for IMD surveillance and audit, describing a range of AFS measures reflecting quality of care for an entire hematology population.

## 2. Methods

### 2.1. Study Design and Setting

The Alfred Hospital is a 638-bed quaternary, university-affiliated, center with trauma, heart/lung transplantation, allogeneic hematopoietic stem cell transplantation (HSCT), cystic fibrosis, burns, hyperbaric medicine and human immunodeficiency virus state-wide services. This was a retrospective observational cohort study from 1 September 2008 to 31 December 2017 coinciding with implementation of our data warehouse which contains imaging and clinical data. Ethics approval was obtained from Alfred Health Office of Ethics (Project no. 104/17) with a waiver of consent granted.

At our center chest CT is performed when IMD is suspected with subsequent bronchoalveolar lavage (BAL) or biopsy performed as tolerated. A diagnostic driven approach using serum galactomannan or aspergillus PCR surveillance is not routine, and when performed these are send-away tests. Empiric antifungal therapy is sometimes commenced in the presence of suspicious radiologic changes while diagnostic investigations continue. All patients have been treated in high efficiency particulate air-filtration rooms since April 2005. Antifungal prophylaxis is prescribed according to institutional protocols noting that intermittent liposomal amphotericin (LAMB) prophylaxis is administered to patients unable to take azole drugs, such as patients on experimental medication enrolled in clinical trials.

An infectious diseases physician and registrar perform regular ward rounds on a referral basis for patients admitted under the hematology service. There is no formal AFS program and the infectious diseases service works in collaboration with hematology and pharmacy services.

### 2.2. Clinical Definitions

IMD was classified by investigators (MAR, DB) according to international definitions [22], with possible infections upgraded to probable or proven categories if a positive microbiological indicator such as isolation of a fungal pathogen or positive galactomannan in bronchoalveolar lavage (BAL) was present [22]. Possible cases lacked positive microbiology but satisfied radiologic and host criteria [22]. In addition, possible cases had documentation in the medical record at the time of diagnosis and/or follow-up (where patients were not palliated), of fungal infection being suspected. If supportive documentation was not found, possible cases were excluded. Our inclusion of possible cases reflects their predominance in real-world practice [15,16] and is consistent with recent AFS guidelines supporting their inclusion [1].

Date of IFD diagnosis was the first day of suspicious radiologic abnormality for patients with possible infection, or positive microbiologic test for patients with probable or proven infection, consistent with previous work [8]. Prolonged neutropenia was defined as absolute neutrophil count <0.5 × 10^9^/L for ≥10 days within 30 days of IMD diagnosis.

Breakthrough-IMD was defined as probable/proven IMD occurring during the course of antifungal prophylaxis or in the 14 days after cessation of antifungal prophylaxis in patients who received at least 7 days of therapy to approximate steady state, reflecting recent expert opinion [25] and other studies [8,14]. Targets for therapeutic drug monitoring (TDM) for antifungal prophylaxis came from our institutional guidelines. Given that treatment of invasive aspergillosis can be over 50 weeks [26], IMD episodes with the same causative fungus in a 12-month period were considered duplicates and excluded, as were episodes with non-filamentous fungi (e.g., *Candida* and *Pneumocystis*) unless co-infection with IMD was present.

Diagnostic delay and its impact on empiric antifungal consumption was determined for a subgroup of probable IMD episodes diagnosed by galactomannan on bronchoalveolar lavage only. Antifungal consumption measured in defined daily dosages (DDD) was evaluated for two intervals, being time to fiberoptic bronchoscopy from diagnostic chest CT scan (interval 1), and the time from bronchoscopy to a galactomannan result reflecting the turnaround time of this send-away test, in interval 2.

### 2.3. Summary of NLP Model Development

NLP is a machine learning based text classifier that aims to improve surveillance of IMD by screening unstructured chest CT reports for the possibility of fungal pneumonia [18,19,20]. NLP generates a probabilistic output ranging from 0%–100% [18,19,20], assigning a binary classification (1 = possible fungal for further review, 0 = not fungal, else) at a threshold of ≥50%. A positive classification triggers review of the medical record to confirm or reject IMD according to international definitions [22]. Development of NLP based on a statistical model was a strenuous task [18,19] involving the manual extraction of 1880 free-text CT reports from 270 patients with IMD and 257 control patients from 3 major Victorian hospitals manually identified from completed mycology studies, clinical databases, pharmacy and microbiology records [18,19]. A subgroup of 449 reports were hand labeled for features of IMD using pre-specified guidelines, at sentence, report, and patient levels [18,19]. The entire report was used for training including reason for the scan, main text, and conclusion. A variety of machine learning, rule-based, and hybrid systems were tested with feature types including bag of words, bag of phrases, and bag of concepts, as well as report-level structured features [18,19]. The best system using support vector machines, achieved a sensitivity over unseen data at a *report* level of 91%, specificity of 79%, receiver operating characteristic area of 0.92, and with few clinically significant missed cases (0.9%), demonstrating its value as a screening tool for IMD [18]. At the *patient* level, performance was outstanding, with NLP identifying 100% of IMD patients (*n* = 188) from a dataset 1247 reports [19] suggesting that if NLP failed to flag the first suggestive CT scan, it would eventually capture the patient on subsequent scans as the radiologist’s language became more conclusive. The statistical model we originally developed [18,19], was changed to a cascaded latent variable model that combined logistic regression and expectation maximization [27] used in this study, that outperformed the earlier models [18,19] and has since progressed to a deep learning-based NLP model that is easier to train [20].

### 2.4. Clinical Data and Audit Methodology

We obtained CT reports from our radiology department from 1 September 2008 to 31 December 2017 belonging to hematology patients. These included any acquisition protocol including the chest such as high-resolution CT, low dose CT, CT pulmonary angiography, and chest CT in combination with any other anatomic site (e.g., brain, neck, abdomen, or pelvis) irrespective of the clinical indication. Reports were screened by NLP [18,19], with positive reports undergoing further review. To determine the rate of false negatives or cases missed by NLP, we (MAR, DB) randomly audited the medical records associated with a convenience sample of 10% negative reports associated with a probability of <50%.

Variables collected included demographics, underlying hematologic disease, HSCT details, histologically proven graft vs. host disease (GVHD), neutropenia, hospitalization duration, exposure to T-cell immunosuppressants, microbiology, histopathology, receipt of antifungal prophylaxis, TDM, invasive diagnostic tests, antifungal drug prescribing, intensive care unit (ICU) admission, and all-cause mortality at 6 or 12 weeks. Annual admissions to the hematology unit were obtained from administrative data but included only multi-day hospitalizations lasting >1 day to determine disease incidence. Data were recorded on a REDcaps database.

### 2.5. Statistical Analysis

Descriptive analyses were based on percentages and frequencies for categorical variables and for continuous variables, means and standard deviations, or medians and interquartile ranges if the data were skewed. Yearly incidence was calculated as the number of IMD cases per 1000 occupied bed-days. A chi-square statistic for trend was used to assess changes in IMD incidence over the study period. A *p* value less than 0.05 was considered statistically significant. Analyses were done using Stata 15 (Stata Corp, College Station TX, USA).

## 3. Results

### 3.1. An Overview of IMD Epidemiology

NLP screened 3014 CT reports flagging 784 reports as possibly fungal for subsequent medical record review (Figure 1). A sample of 10 anonymized reports with NLP probabilities and binary outputs is shown in Appendix A. Following medical record review and adjudication using international criteria [22], we identified 205 IMD episodes across 50,303 admissions. Among the 185 patients with IMD, 68% were male with median age 58 years (IQR 44-67); 30% were elderly (i.e., 65 years or more) and 49% of episodes were associated with active hematological disease (Table 1).

Among non-HSCT episodes, we noted a predilection for patients with acute myeloid leukemia (AML, 67%), acute lymphoblastic leukemia (ALL, 12%) and non-Hodgkin’s lymphoma (NHL, 7.3%). Enrolment in a clinical trial was present in 20% of IMD episodes, with AML the primary condition in 83%. Among the 33% of HSCT recipients, 78% were allogeneic with 35% (18/52) occurring early (i.e., ≤100 days) post-transplant. Periods of risk represented by median onset of IMD from diagnosis of hematological disease varied by hematological condition and was substantially later in onset for patients with multiple myeloma and chronic lymphocytic leukemia (Appendix A). Yearly incidence of IMD decreased over the study period from 2.85 per 1000 occupied bed-days in 2009 to 1.60 in 2017 (test for trend, *p* = 0.006, Appendix A).

IMD episodes were possible and probable/proven in 56% and 44%, respectively (Table 2). *Aspergillus* species accounted for the majority (73%) of probable/proven episodes followed by the *Mucorales* at 18% with non-*Aspergillus* molds accounting for 31% of episodes. The number of non-*Aspergillus* molds ranged from 1 to 8 per year (mode = 3), with incidence remaining fairly steady over the 9-year study period (test for trend, *p* = 0.057).

Random audit of 10% of reports (*n* = 223) screened negative by NLP, revealed 2 patients with possible IMD who should have been flagged positive but were missed for unclear reasons, resulting in an overall miss rate of 0.9% (2/223) (Appendix A). There were 2 cases (2009, 2011) from the training dataset from Alfred Health (2003–2011) used to develop the NLP model that also appeared in this study.

### 3.2. Process and Outcome Measures: Breakthrough-IMD, TDM, Missed Prophylaxis Opportunities

Breakthrough probable/proven IMD on antifungal prophylaxis occurred in 60% of probable/proven cases (*n* = 53/88) (Table 3).

The majority (89%) of breakthrough-IMD was associated with mold-active prophylaxis including posaconazole (*n* = 15, 28%), intermittent LAMB prophylaxis (*n* = 17, 32%), voriconazole (*n* = 13, 25%), and caspofungin (*n* = 2, 3.8%), while 11% (*n* = 6) complicated fluconazole prophylaxis (Figure 2). Breakthrough-probable/proven IMD episodes complicating intermittent LAMB prophylaxis included 6 ALL patients receiving induction chemotherapy, and a further 8 AML patients where 5 were enrolled in clinical trials that contraindicated azole use.

For those breakthrough-probable/proven IMD cases receiving posaconazole or voriconazole prophylaxis, TDM was performed in the 14 days prior to IMD diagnosis in (53%) 8/15 of posaconazole and 9/13 (69%) of voriconazole recipients and was therapeutic according to our institutional guidelines in 63% and 78% of recipients, respectively.

### 3.3. Effect of Diagnostic Delays on Antifungal Consumption

An invasive diagnostic test was performed in 83% (170, 170/205) of episodes, with fiberoptic bronchoscopy the most common procedure (60%, 123/205) (Table 3) that increased over the study period (Appendix A). The median time from CT scan to fiberoptic bronchoscopy was 2 days but was variable with 54% occurring within 2 days, 32% occurring between 3 to 5 days and 14% occurring 5 days or more (Table 3). The average time of galactomannan test turnaround was 12 days (range 7–22 days) for 24 IMD episodes associated with a positive bronchoalveolar lavage galactomannan.

A subgroup analysis of 19 probable IMD episodes diagnosed solely by a positive bronchoalveolar galactomannan revealed a high median empiric antifungal consumption in interval 2 associated with the long turnaround time of the galactomannan result, dominated by voriconazole at 14 DDDs and followed by LAMB at 10.5 DDDs per episode, as shown in Appendix A.

### 3.4. Probable/Proven IMD in Patients Who Did Not Receive Prophylaxis

A variety of patients developed probable/proven IMD (*n* = 35 episodes) who did not receive antifungal prophylaxis in the 14 days prior to IMD diagnosis (Appendix A). Non-transplant groups had either newly diagnosed or refractory disease. Among allogeneic-HSCT recipients, three probable/proven IMD episodes occurred early post-transplant (i.e., at 33, 40, 64 days), and 6 episodes complicated histologically proven chronic GVHD that was present in the 60 days prior to IMD diagnosis, representing potential missed prophylaxis opportunities (Table 3). 

### 3.5. Clinical Outcomes

Resource utilization was high (Table 2) with ICU admission required in 26% of episodes, of whom 32% required invasive ventilation. Of 130 evaluable episodes, 6 and 12 week all-cause mortality was 40% and 52%, respectively.

## 4. Discussion

We report the first successful use of applied machine learning for institutional case-finding and audit of IMD in an entire hematology population. This approach provided detailed patient-level data which is fundamental to AFS but rarely achieved in practice [6]. We were able to characterize our highest risk as well as low risk/emerging patient groups, breakthrough infections on a range of prophylactic antifungal drugs and missed prophylaxis opportunities. Our semi-automated approach to surveillance and audit still required manual chart review but made the finding of a rare infection feasible. This is significant considering we identified on average 22 incident IMD episodes from over 5000 admissions annually to the hematology unit, which would not have been possible without this technology. We were also able to establish a baseline for outbreak detection, describe reproducible metrics like the impact of diagnostic delays on empiric antifungal consumption and apply figures to benchmark recommendations for optimum patient care in AFS [7] beyond the commonly reported antifungal consumption and cost [6]. This targeted review of process and outcome measures for each IMD episode provided a rich overview of our fungal epidemiology and AFS practice, laying the groundwork for future quality improvement work.

Missed opportunities and variability from institutional guidelines underscore the importance of regular clinical audit. There were three patients in the early (within 100 days) post allogeneic-HSCT period who qualified for prophylaxis but did not receive it in the previous 14 days, in addition to 6 HSCT recipients with concurrent histologically-proven GVHD where prophylaxis may have been indicated. Of concern, 60% of the 88 evaluable episodes with probable/proven IMD represented breakthrough-IMD on antifungal prophylaxis when applying a definition concordant with recent expert opinion [25]. Breakthrough-IMD also occurred in a variety of conditions where a standardized approach is lacking [29]. For example, 32% (*n* = 17/53 episodes) of probable/proven breakthrough-IMD complicated intermittent LAMB prophylaxis. These included 6 episodes during induction chemotherapy for ALL, where interactions between azole drugs and vinca alkaloids are well recognized [30], and a further 5 episodes in AML patients on experimental drugs contraindicating azole use. Although posaconazole TDM is contentious [31], it is advocated in our setting, but like voriconazole TDM, was performed inconsistently in the 14 days prior to breakthrough probable/proven IMD in those patients receiving prophylaxis.

An increase in the performance of fiberoptic bronchoscopy over the study period is encouraging but room for improvement remains, especially given our high incidence of non-*Aspergillus* molds which comprised 31% of probable/proven episodes. Only 54% of fiberoptic bronchoscopies were performed within 2 days of CT scan, which is suboptimal considering that the diagnostic yield declines after 4 days from symptom (not radiological) onset [5]. We identified potential cost savings by quantifying the impact of diagnostic delays on empiric antifungal consumption during (1) time from CT scan to bronchoscopy and (2) galactomannan test turnaround, in a subgroup of 19 probable IMD episodes where bronchoalveolar lavage galactomannan established the diagnosis. The mean turnaround time for a send-away galactomannan result of 12 days was associated with high liposomal amphotericin consumption which has clear implications on cost, especially as a negative galactomannan usually results in streamlining or cessation of empiric antifungal drugs in our setting. This data represents a baseline from which improvements in bronchoscopy timeliness and on-site rapid diagnostics can be argued, but building a business case will rely on AFS teams presenting locally relevant data as we have done.

Our study has several limitations, including its retrospective nature and single center focus. While our findings may not be extrapolated to other centers, the methodology of NLP assisted surveillance and audit is generalizable. The *in silico* validation of our NLP model described here is the step prior to real-world validation and implementation of applied artificial intelligence [32]. We are progressing through those stages with real-world multi-center prospective validation of our deep learning-based NLP model underway (ClinicalTrials.gov NCT03793231). The combination of microbiology and antifungal drug data with NLP predictions in our expert system [24] does decrease false positives, but is challenging to implement due to the cost of integrating legacy systems in hospitals. NLP is tuned for fungal pneumonia, and may miss extrapulmonary presentations like fungaemia or sinusitis. Finally, NLP was associated with a low miss rate of 0.9% but the reason for this failure is unclear, pointing to the “black-box”-like inscrutability of artificial intelligence [32].

## 5. Conclusions

Surveillance and audit of IMD is fundamental to AFS programs [1,6,7] but is currently inefficient and restrictive. Valerio et al. recommends that AFS programs should have measurable indicators that are prospectively recorded and simple to obtain, allowing AFS teams to focus on intervention rather than data collection [33]. We would add that these indicators should be patient-level and reflect quality of care, rather than fluctuations in drug consumption or cost. NLP is a valuable tool in the AFS armamentarium that represents a potentially scalable solution to the challenge of surveillance and audit, that leverages unstructured radiology data which is a readily available resource in hospitals.

## Figures and Tables

**Figure 1 jcm-08-01390-f001:**
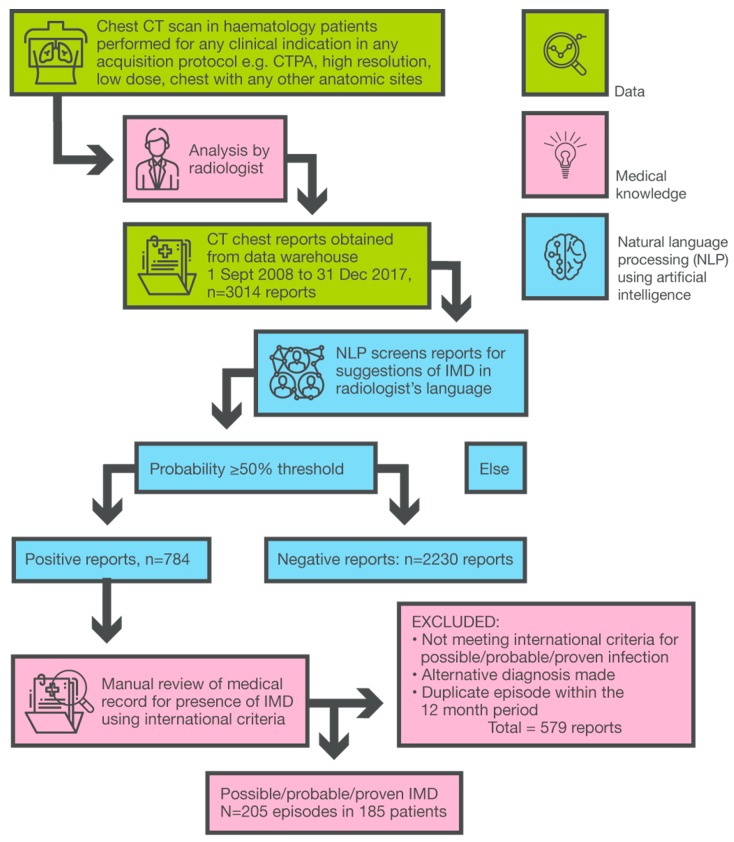
IMD case-finding and audit using natural language processing of chest CT reports in hematology patients. CT, computed tomography; IMD, invasive mold disease; CTPA, CT-pulmonary angiography.

**Figure 2 jcm-08-01390-f002:**
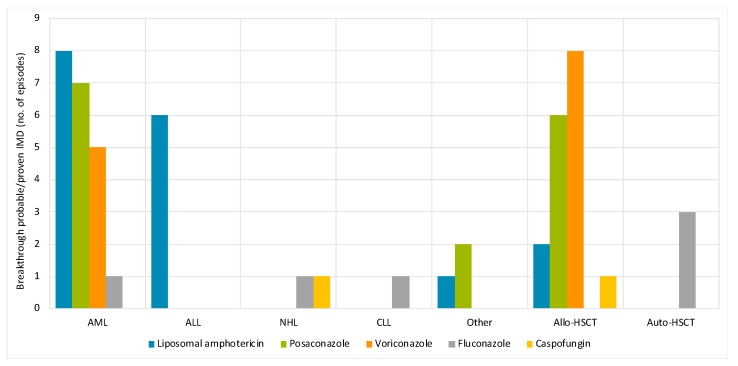
Breakthrough probable/proven IMD (*n* = 53 episodes) on antifungal prophylaxis. Abbreviations: IMD, invasive mold disease; AML, acute myeloid leukemia; ALL, acute lymphoblastic leukemia; NHL, non-Hodgkin’s lymphoma; CLL, chronic lymphocytic leukemia; HSCT, hemopoietic stem cell transplant (allo-, allogeneic; auto-autologous). Other refers to primary blastic dendritic cell neoplasm, blastic plasmacytoid dendritic cell neoplasm, mixed phenotype acute leukemia.

**Table 1 jcm-08-01390-t001:** Characteristics of patients with invasive mold disease.

Characteristics	N (%)
Number of patients	185
Age, years	
Median [IQR]	58 (44–67)
Elderly (≥65 years)	56 (30)
Male gender	125 (68)
Number of IMD episodes	205
Outpatients	14 (6.8)
Length of hospitalization, days	
Median [IQR]	29 (15–43)
Underlying hematologic disease, *n* = 138 non-HSCT episodes	
Acute myeloid leukemia	93 (67)
Acute lymphoblastic leukemia	17 (12)
Non-Hodgkin lymphoma	10 (7.3)
Myelodysplastic syndrome	5 (3.6)
Multiple myeloma	3 (2.2)
Chronic lymphocytic leukemia	3 (2.2)
Hodgkin lymphoma	2 (1.5)
Acute promyelocytic leukemia	1 (0.8)
Other ^a^	4 (2.9)
HSCT recipients	67 (33)
Type of HSCT	
Autologous	15 (22)
Allogeneic	52 (78)
Matched related	35 (52)
Unrelated	16 (24)
Mismatched	1 (1.5)
Early post allo-HSCT (≤100 days)	18 (35%)
Late post allo-HSCT (>100 days)	34 (65%)
Stem cell source in allogeneic-HSCT recipients, *n* = 52 episodes	
Peripheral blood	48 (94)
Cord blood	2 (3.9)
Marrow	1 (2.0)
Status of hematological disease, *n* = 203 episodes ^b^	
Newly diagnosed or receiving first induction therapy	49 (24)
Complete remission	55 (27)
Active hematologic disease (partial remission, progression, relapsed/refractory)	99 (49)
Enrolled in clinical trial	42 (20)
Neutropenia ≤0.5 × 10^9^/L lasting ≥10 days in 30 days prior to IMD diagnosis	108 (53)
Duration of neutropenia ≤0.5 × 10^9^/L, median [IQR]	22 (15–37)

^a^ Includes 1 patient with acute leukemia of ambiguous lineage, two mixed phenotype acute leukemia, 1 each of blastic dendritic cell neoplasm and blastic plasmacytoid dendritic cell neoplasm. ^b^ Non-malignant conditions (*n* = 2) excluded. Abbreviations: IMD, invasive mold disease; IQR, interquartile range; HSCT, hemopoietic stem cell transplant.

**Table 2 jcm-08-01390-t002:** Characteristics of IMD episodes.

IMD Characteristics, *n* = 205 Episodes	N (%)
EORTC/MSG classification	
Proven/probable	90 (44)
Possible	115 (56)
Site of infection	
Localized	190 (93)
Disseminated	15 (7.3)
Site of involvement	
Lung only	198 (97)
Sinus	5 (2.4)
Bloodstream	4 (2.0)
Liver	2 (1.0)
Spleen	2 (1.0)
CNS	2 (1.0)
Other ^a^	4 (2.0)
Organism in probable/proven IMD episodes, *n* = 90 episodes ^b^	
*Aspergillus* species	66 (73)
*Aspergillus* fumigatus	20 (30)
Non-fumigatus *Aspergillus* species	9 (14)
*Aspergillus* species not otherwise specified	25 (38)
Positive galactomannan from BAL	24 (27)
Non-*aspergillus* molds	31 (34)
*Mucorales*	16 (18)
*Lomentospora prolificans*	5 (5.6)
Other rare molds ^c^	10 (11)
Mixed fungal infections	8 (8.9)
*Aspergillus* PCR positive ^d^	13 (14)
Panfungal PCR positive ^d^	5 (5.6)
Possible IMD episodes	115
*Aspergillus* PCR positive	12 (10)
Panfungal PCR positive	5 (4.3)
Clinical outcomes	
ICU admission	53 (26)
Invasive ventilation	17 (32)
All-cause mortality, *n* = 130 evaluable episodes	
6 week	52 (40)
12 week	68 (52)

^a^ Other refers to colon (*n* = 1), heart and great vessels (*n* = 1), larynx (*n* = 1), pancreas and kidneys (*n* = 1). ^b^ More than one species in some episodes. ^c^
*Paecilomyces* spp. (*n* = 2), *Phialemonium* spp. (*n* = 1), *Penicillium* spp. (*n* = 1), *Fusarium* spp. = 2, *Acremonium* spp. (*n* = 2), *Alternaria alternata* (*n* = 1), *Saccharomyces* (*n* = 1). ^d^ Aspergillus PCR was positive on bronchoalveolar lavage specimens; Panfungal PCR was positive on 3 tissue specimens (1 sinus, 2 lung) and 2 bronchoalveolar lavage specimens. Abbreviations: IMD, invasive mold disease; EORTC/MSG, European Organization for Research and Treatment of Cancer/Invasive Fungal Infections Cooperative Group and the National Institute of Allergy and Infectious Diseases Mycoses Study Group (EORTC/MSG ^22^; PCR, polymerase chain reaction; ICU, intensive care unit; IQR, interquartile range.

**Table 3 jcm-08-01390-t003:** Gaps and opportunities for antifungal stewardship.

Characteristics	N (%)
Breakthrough probable/proven IMD ^a^	53/88 (60)
Posaconazole	15 (28)
Liposomal amphotericin B	17 (32)
Voriconazole	13 (25)
Fluconazole	6 (11)
Caspofungin	2 (3.8)
TDM performed in 2 weeks prior to breakthrough probable/proven IMD diagnosis ^b^	
Posaconazole prophylaxis	8/15 (53)
Therapeutic	5 (63)
Voriconazole prophylaxis	9/13 (69)
Therapeutic	7 (78)
Invasive diagnostic tests, (*n* = 205 episodes)	170 (83)
Fiberoptic bronchoscopy	123 (60)
Lung	27 (13)
Sinus	8 (3.9)
Other site	9 (4.3)
Lung resection	3 (1.4)
Interval from CT to fiberoptic bronchoscopy, *n* = 117 episodes ^c^	
Median time [IQR], days	2 (1–5)
≤2 days	63 (54)
3–5 days	38 (32)
>5 days	16 (14)
Time from bronchoalveolar lavage galactomannan request to result, *n* = 24 episodes, mean (range), days	12 (7–22)
Potential missed opportunities for antifungal prophylaxis in probable/proven IMD episodes ^d^	
Early post allogeneic-HSCT (≤100 days)	3/18 (17)
Chronic GVHD ^e^ in 60 days prior to IMD diagnosis	6/13 (46)
Neutrophils ≤0.5 × 10^9^/L for >3 weeks and <5 weeks	9/27 (33)
Neutrophils ≤0.5 × 10^9^/L for >5 weeks	9/28 (32)

^a^ Breakthrough IMD defined as onset during antifungal prophylaxis and up to 14 days after cessation in patients who received at least 7 days of consecutive antifungal therapy. ^b^ Trough levels were considered therapeutic if >0.7 mg/L and >1.0 mg/L for posaconazole and voriconazole respectively according to institutional guidelines. ^c^ Defined as the time from computed tomography (CT) scan to fiberoptic bronchoscopy, with diagnostic CT scan assumed to be immediately prior to fiberoptic bronchoscopy. Bronchoscopy not prompted by an abnormal CT scan excluded (*n* = 6 episodes). ^d^ High to intermediate risk factors for IMD in guidelines [28]. ^e^ Histologically proven GVHD. Abbreviations: IMD, invasive mold disease; TDM, therapeutic drug monitoring; IQR, interquartile range; HSCT, hemopoietic stem cell transplant; GVHD, graft vs. host disease.

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
