# Peer review of "Closing the Gap in Surveillance and Audit of Invasive Mold Diseases for Antifungal Stewardship Using Machine Learning"

_jcm, 2019, doi:10.3390/jcm8091390_

Round 1

Reviewer 1 Report

The authors have submitted a well-written study in which a semi-automated process using natural language processing was applied to CT chest radiology reports to identify cases of suspected invasive mold disease. The challenge of surveillance and audit of invasive mold diseases is clearly described.

This is an interesting and novel application of machine learning to address a difficult problem in clinical care of immunosuppressed patients. The authors describe how their algorithm performed with respect to identifying possible cases of IMD based on radiology reports and note a 0.9% false negative rate. One of the limitations of this study is that it uses radiology reports as the ground truth. The authors note that at a patient level, NLP identified 100% of IMD patients. How did the radiologists perform? Were any cases missed by the radiologists, which would thus be missed by NLP that depends on radiology reports?  As the authors note, this is a great application for deep neural networks, which could more objectively assess the likelihood of IMD without the subjectivity introduced by radiologists’ interpretation.

Overally, this is a well-executed study and I recommend it for publication after minor clarifications.

Author Response

We thank the reviewer for their helpful comments.

Reviewer query:

One of the limitations of this study is that it uses radiology reports as the ground truth. The authors note that at a patient level, NLP identified 100% of IMD patients. How did the radiologists perform? Were any cases missed by the radiologists, which would thus be missed by NLP that depends on radiology reports?  

Response: Ground truth or the reference standard used to originally train the text classifier (Martinez et al J Biomed Inform 2015, Ananda-Rajah et al Plos One 2014) were cases adjudicated by clinician experts using EORTC/MSG (2008) criteria. These cases came from datasets belonging to previously published mycology studies, infectious diseases consultation databases and after reviewing pharmacy dispensing records for antifungals. The 100% sensitivity of NLP (Martinez et al J Biomed Inform 2015) refers to the machine learning text classifier correctly identifying the 188 patients with IFD adjudicated by EORTC/MSG criteria from a cohort of 1247 reports. Ground truth is not the radiologist opinion but clinical, host and microbiological data as per consensus guidelines.

Cases missed by NLP at report level, did occur in the held out validation set (0.9%) (Ananda-Rajah et al Plos One 2014) but the negative impact of these missed cases was minimised by NLP detecting subsequent reports when the radiologist language usually became more definitive making patient level (i.e including index and follow up scans) very good.

Reviewer 2 Report

Ms. Ref. No.: jcm-566108

Title: Closing the gap in surveillance and audit of invasive  mold diseases for antifungal stewardship using  machine learning

Overview and general recommendation:

The authors reported the first successful use of applied machine learning for institutional case-finding and audit of IMD in an entire haematology population. They characterized highest risk as well as low risk/emerging patient groups, breakthrough infections on a range of prophylactic antifungal drugs and missed prophylaxis opportunities. Random audit of 10% negative reports revealed two clinically significant misses (0.9%, 2/223). This suggests that NLP is an inclusive and potentially scalable screening technology for IMD, that facilitates surveillance and audit, relevant to AFS.

Minor Comments:

The abstract needs to be rewritten to clearly state the advantage of NLP and the significance of the study. Authors also may need to describe their aim and objective in the introduction part. Figure 2: Remove the numbers on the bars as they are the same as represented on the Y-axis.

Author Response

We thank the reviewer for their helpful comments.

The abstract has been revised.

Clinical audit of invasive mold disease (IMD) in haematology patients is inefficient due to the difficulties of case finding. This results in antifungal stewardship (AFS) programs preferentially reporting drug cost and consumption rather than measures that actually reflect quality of care. We used machine learning-based natural language processing (NLP) to non-selectively screen chest tomography (CT) reports for pulmonary IMD, verified by clinical review against international definitions and benchmarked against key AFS measures. NLP screened 3014 reports from1 September 2008 to 31 December 2017), generating 784 positives that after review, identified 205 IMD episodes (44% probable-proven) in 185 patients from 50,303 admissions. Breakthrough-probable/proven-IMD on antifungal prophylaxis accounted for 60% of episodes with serum monitoring of voriconazole or posaconazole in the 2 weeks prior performed in only 53% and 69% respectively. Fibreoptic bronchoscopy within 2 days of CT scan occurred in only 54%. The average turnaround of send-away bronchoalveolar galactomannan of 12 days (range 7-22) was associated with high empiric liposomal amphotericin consumption. Random audit of 10% negative reports revealed two clinically significant misses (0.9%, 2/223). This is the first successful use of applied machine learning for institutional IMD surveillance across an entire haematology population describing process and outcome measures relevant to AFS. Compared to current methods of clinical audit, semi-automated surveillance using NLP is more efficient and inclusive by avoiding restriction based on underlying haematologic condition, and has the added advantage of being potentially scalable.

The aim of the study has been included in the introduction.

The aim of this study was to characterize institutional IMD burden along with process and outcome measures relevant to AFS in order to identify areas for improvement. What follows is a clinical (rather than technical) study of applied machine learning that represents an in silicoanalysis of NLP for IMD surveillance and audit, describing a range of AFS measures reflecting quality of care for anentirehaematology population.

Numbers have been removed from Figure 2.

Reviewer 3 Report

Baggio and colleagues describe an audit using machine learning.

The paper is well written and easy to follow.

There are some issues that need to be considered as essentially the machine learning is a case-finding tool.

NLP:

a) does this take into account history given on the request form?

b) Radiologists generally write conclusions based on questions asked therefore if IFI is queried -> conclusion will generally contain "fungal infection". Does the NLP take this into account or only examine the findings

c) Why is the False positive rate high at 78% especially the probability is set at 50%; expect a 50% FP rate?

d) Line 103-107 this seems some what contradictory - why exclude cases if the radiology meets criteria? 

Table2

Would revise and present fungal data by patient rather than aggregate data. PCR tests - specimen type should be noted. 

Table 3

Would revise and present the data in the negative (i.e. numbers of opportunities lost)

What proportion of patients should not have been on prophylaxis based on guidelines?

Invasive diagnostic tests how is the system able to control this?

Line 294 5000 admissions ? CT reports were only examined = 2230+784 in figure 1.

Line 306 - Prophylaxis is not absolute - breakthrough is therefore suspected and will be the standard pathogens. However, "breakthrough" in the setting of sub-therapeutic levels is a drug exposure issue and I would argue not a breakthrough as described. Would modify this import distinction throughout the text.

Authors should explain why a case-finding NLP based on a patients EMR would not result in the same conclusions.

Author Response

We thank the reviewer for their helpful comments and have responded to queries below.

NLP:

a) does this take into account history given on the request form?

Yes, the machine learning classifier is trained on the entire report. A separate analysis of the reason for the scan being performed revealed that the language is neutral regarding fungal suspicion (not published).

b) Radiologists generally write conclusions based on questions asked therefore if IFI is queried -> conclusion will generally contain "fungal infection". Does the NLP take this into account or only examine the findings.

The labelled data is central to understanding how NLP is biased to flag suspicous reports and this labelling process is extensively described in Martinez et al (J Biomed Inform 2015) available on the fungalAi website. The whole report was used for training. Each sentence was labelled using pre-specified criteria which took into account any specific mention of fungal infection by either clinician or radiologist.

The following has been added: line 137 “The entire report was used for training including reason for the scan, main text and conclusion. “

c) Why is the False positive rate high at 78% especially the probability is set at 50%; expect a 50% FP rate?

The false positive rate is high because the text classifier is tuned for a high sensitivity in order to avoid missing these uncommon cases. A probability of 50% refers to a cut off for the prediction where a prediction of <50% or 50%+ results in a report being classified as either negative or positive respectively which is in keeping with the way the text classifier was developed (Martinez et al, Ananda-Rajah et al). With this cut off the ROC for IFD detection was 0.92 when the classifier was developed (Ananda-Rajah et al Plos One 2014).

d) Line 103-107 this seems some what contradictory - why exclude cases if the radiology meets criteria? 

Supportive documentation of treatment or suspicion of  fungal infection was required for all possible cases in addition to the EORTC/MSG (2008) criteria. This was done to strengthen the study and avoid including cases with suggestive radiology but no evidence that clinicians treating the patients actually thought they had IFD. In doing so, approximately 40 cases were excluded from the study. We stand by this decision as it increases the likelihood that these possible cases were actually IFD rather than alternative conditions as it reflects the thoughts of the treating clinicians at the time.

Table2

Would revise and present fungal data by patient rather than aggregate data. PCR tests - specimen type should be noted. 

A footnote has been added to the table for PCR tests.

We do not believe that revision will aid additional interpretation. Several patients had more than one IFD during the study period. Some studies (Ceesay Br J H 2015) report as per IFD episode as we have done rather than as per patient. This issue was not raised by the other reviewers, so we would prefer to retain the information as is.

Table 3

Would revise and present the data in the negative (i.e. numbers of opportunities lost)

We would seek clarification on this point, it is not clear what the reviewer suggests. It is unclear if presenting data as opportunities lost will make interpretation easier for the reader. 

What proportion of patients should not have been on prophylaxis based on guidelines?

This is a good question but one that can not be answered because we did not evaluate concordance with our institutional guidelines which have likely changed a few times over this study period. Appropriateness of prescribing  according to guidelines was not examined in this study but will be examined in future work especially for intermittent LAMB prophylaxis which this study highlighted as a potential problem.

Invasive diagnostic tests how is the system able to control this?

This study is a targeted review of process measures (like invasive diagnostics, TDM) associated with IFD to identify areas of improvement. The system does not control nor influence invasive tests-these are performed routinely.

Line 294 5000 admissions ? CT reports were only examined = 2230+784 in figure 1.

This is correct. There are on average 5000 admissions per year inclusive of day only and multi-day admissions (in 2017, 1237 multiday admissions from a total of 7298 admissions per year) but approximately 250-350 CT scans are performed per year. The majority of admissions are single day admissions associated with attendance for chemotherapy.

Line 306 - Prophylaxis is not absolute - breakthrough is therefore suspected and will be the standard pathogens. However, "breakthrough" in the setting of sub-therapeutic levels is a drug exposure issue and I would argue not a breakthrough as described. Would modify this import distinction throughout the text.

We understand the reviewer’s point of view but would disagree. The definition for breakthrough IFD used is similar to Lerolle et al (2014) and Ananda-Rajah et al (2012) who did not account for sub therapeutic levels. Recent consensus recommendations specific to breakthrough IFD (Cornely et al Mycoses 2019)  do not take sub therapeutic levels into account when defining break through IFD. Arguably this makes definitions more reproducible as levels are often done ad hoc in practice and TDM for posaconazole is controversial.  We included levels where available for posaconazole and voriconazole  because many clinicians believe they are important and by presenting this data, they can judge for themselves.

Authors should explain why a case-finding NLP based on a patients EMR would not result in the same conclusions.

The possibility of applying NLP on a patients EMR being equally useful for case finding is purely speculative and we would avoid making any conclusions. NLP for IFD case finding using  the entire EMR has not been developed, published nor is it undergoing real world validation. It is possible that FHIR may assist with disease surveillance but again this is speculative. Importantly, the EMR is not present in all hospitals but major centres in high and middle income centres have radiology PACS systems which can be used now.